# The Role of GRP and MGP in the Development of Non-Hemorrhagic VKCFD1 Phenotypes

**DOI:** 10.3390/ijms23020798

**Published:** 2022-01-12

**Authors:** Suvoshree Ghosh, Johannes Oldenburg, Katrin J. Czogalla-Nitsche

**Affiliations:** 1Institute of Experimental Haematology and Transfusion Medicine, Venusberg Campus 1, University Clinic Bonn, 53127 Bonn, Germany; Suvoshree.Ghosh@ukbonn.de (S.G.); Johannes.Oldenburg@ukbonn.de (J.O.); 2Center for Rare Diseases Bonn, Venusberg Campus 1, University Clinic Bonn, 53127 Bonn, Germany

**Keywords:** VKCFD1, GGCX, GRP/UCMA, MGP

## Abstract

Vitamin K dependent coagulation factor deficiency type 1 (VKCFD1) is a rare hereditary bleeding disorder caused by mutations in γ-Glutamyl carboxylase (*GGCX*) gene. The GGCX enzyme catalyzes the γ-carboxylation of 15 different vitamin K dependent (VKD) proteins, which have function in blood coagulation, calcification, and cell signaling. Therefore, in addition to bleedings, some VKCFD1 patients develop diverse non-hemorrhagic phenotypes such as skin hyper-laxity, skeletal dysmorphologies, and/or cardiac defects. Recent studies showed that *GGCX* mutations differentially effect γ-carboxylation of VKD proteins, where clotting factors are sufficiently γ-carboxylated, but not certain non-hemostatic VKD proteins. This could be one reason for the development of diverse phenotypes. The major manifestation of non-hemorrhagic phenotypes in VKCFD1 patients are mineralization defects. Therefore, the mechanism of regulation of calcification by specific VKD proteins as matrix Gla protein (MGP) and Gla-rich protein (GRP) in physiological and pathological conditions is of high interest. This will also help to understand the patho-mechanism of VKCFD1 phenotypes and to deduce new treatment strategies. In the present review article, we have summarized the recent findings on the function of GRP and MGP and how these proteins influence the development of non-hemorrhagic phenotypes in VKCFD1 patients.

## 1. Introduction

γ-Glutamyl carboxylase (GGCX) is a post translational modifying enzyme and the only one that γ-carboxylates vitamin K dependent (VKD) proteins at specific glutamic acid (Glu) residues to γ-carboxyglutamic acid residues (Gla) [1,2,3]. This process takes place in the membrane of the endoplasmic reticulum (ER), which requires vitamin K hydroquinone (KH_2_), carbon dioxide (CO_2_), and oxygen (O_2_) as co-factors. During γ-carboxylation, the cofactor KH_2_ is oxidized to vitamin K epoxide (K > O) by the epoxidase activity of GGCX. K > O is recycled back to KH_2_ by the enzyme vitamin K 2,3-epoxide reductase complex subunit 1 (VKORC1) [4,5]. VKD proteins include specific clotting factors essential for blood coagulation as (F) FII, FVII, FIX, FX, and Protein C (PC), S, and Z. Further eight non-hemostatic VKD proteins are known, matrix Gla protein (MGP), osteocalcin (BGLAP), upper zone of growth plate and cartilage matrix associated protein (UCMA/GRP), growth arrest specific 6 (GAS6) proline-rich Gla proteins (PRGPs) 1 and 2, and transmembrane Gla proteins (TMGs) 3 and 4 [6]. These proteins have diverse functions such as in mineralization or cell signaling. VKD proteins are composed of different domains namely the signal peptide, the propeptide domain, the Gla domain, and additional protein specific functional domains. The signal peptide and the propeptide are cleaved in the ER and Golgi apparatus, respectively. The Glu residues that undergo γ-carboxylation by GGCX are located in the Gla domain of VKD proteins [6]. After additional modifications as for example N-linked glycosylation or phosphorylation, the mature VKD proteins are mostly secreted except for PRGP1-2 and TMG 3-4, which are transported to the membrane.

Mutations in *GGCX* gene cause a rare hereditary bleeding disorder called Vitamin K dependent coagulation factor deficiency type 1 (VKCFD1; OMIM #277450) [3,7,8,9,10,11,12,13,14,15,16,17]. VKCFD1 patients are primarily diagnosed by decreased activity of all VKD clotting factors. Due to the reduced clotting factor activities, the patients suffer from spontaneous bleedings or extended bleedings after surgery as tooth extraction, and/or menorrhagia. High dose treatment with vitamin K_1_ reverses the bleeding phenotype in most cases (10 mg phylloquinone per day). However, VKCFD1 patients with certain genotypes will not achieve normal clotting factor activities with only vitamin K treatment. Therefore, patients with such severe genotypes should be treated with Prothrombin complex concentrate especially during surgery [18]. Mutations in GGCX also cause mineralization phenotypes in skin such as Pseudoxanthoma elasticum (PXE)-like phenotype, and/or cardiac abnormalities, and/or skeletal defects in several patients but interestingly not in all [19]. These additional phenotypes reported in VKCFD1 patients are most likely due to the under-carboxylation of one or more non-hemostatic VKD proteins caused by the insufficient γ-carboxylase activity of the mutated GGCX protein. Previous studies have shown that regulation of VKD proteins and vitamin K availability has an impact on maintaining physiology of the skeletal and cardiac system [20,21,22,23,24].

In this review, we discuss recent findings on MGP and GRP as regulators of mineralization and their impact on the development of skeletal defects, the skin hyper-laxity, and the cardiac abnormalities in VKCFD1 patients.

## 2. VKD Calcification Inhibitors

Physiological calcification is crucial for maintaining tissue homeostasis, which is depending on the proper regulation of both pro and anti-mineralization factors [25]. The imbalance in mineralization causes ectopic calcification leading to diverse pathological phenotypes. Two VKD proteins, MGP and GRP, are known to function, as calcification inhibitors and under-carboxylation of these proteins are associated with pathological calcification.

### 2.1. Matrix Gla Protein

MGP is highly expressed in the extracellular matrix of calcified tissues, in cartilages and in vascular smooth muscle cells (VSMC) [26,27,28]. It is composed of 84 amino acid (aa) residues, where five specific glutamic acid residues (21, 56, 60, 67, 71 aa) undergo γ-carboxylation. In addition, three serine residues (22, 25, 28 aa) undergo phosphorylation. These post-translational modifications co-ordinates the binding of MGP with calcium crystals and bone morphogenetic protein (BMP) that assists in inhibiting calcification [29]. Schurgers et al. have reviewed the role of MGP in ectopic calcification and have narrowed down different mechanism of MGP’s inhibitory function, i.e., by inhibition of calcium-phosphate precipitation, inhibition of trans-differentiation of VSMC, and regulation by VSMC derived apoptotic bodies’ and matrix vesicles [30]. The osteogenic trans-differentiation of VSMC is regulated by MGP’s interaction with BMP-2. MGP can bind and inactivate BMP-2 via its Gla domain, which is an osteogenic growth factor that induces bone and cartilage development [31,32,33].

Major evidence of an inhibitory role of MGP in soft tissues comes from the knockout mouse model. *Mgp*^−/−^ mice were born normal with regular mendelian frequency but died after two months due to extensive arterial calcification that led to blood vessel rupture [34]. In *Mgp*^−/−^ mice chondrocyte-like cells replaced the medial smooth muscle cells, and calcified matrix was deposited in the artery [35]. Further, Marulanda et al. reported the development of midfacial hypoplasia in *Mgp*^−/−^ mice [36]. These mice showed a shorter and abnormally mineralized nasal septum. A rescue was seen with transgenic restoration of MGP expression in chondrocytes, which fully corrected the craniofacial anomalies.

The importance of MGP function has been elucidated in humans with the identification of mutations in *MGP* which is associated with an autosomal recessive disorder called Keutel syndrome. Keutel syndrome patients are characterized mainly by abnormal cartilage calcification and midfacial hypoplasia [37]. Forty-two cases of Keutel syndrome have been reported until now, where only 30% of them were genotyped for MGP, by which eight different loss-of-function mutations were identified in the MGP gene [38]. MGP is the only protein among all other non-hemostatic VKD proteins, which is associated with a monogenic disorder so far.

Apart from the vascular and osseous function, MGP was shown to have a role in sperm maturation. Ma et al. showed that sperm maturation is promoted by intercellular calcium signaling, which is regulated by γ-carboxylated MGP [39].

Altogether, these studies show that MGP has a major role in binding of calcium ions in the vasculature, cartilage tissue, and sperm, where absence of MGP lead to calcified or underdeveloped tissue.

### 2.2. Gla Rich Protein

GRP is the most recent identified VKD protein. It is highly expressed in osseous and soft tissues as for example skin. GRP has a total of 138 aa residues and after the cleavage of the signal peptide and propeptide, the 74 aa mature protein is released into the circulation. It has the highest number of Gla residues of all known VKD proteins (15 in humans). Therefore, it was named Gla rich protein. In 2008, Viegas et al. were the first to isolate this novel vitamin K dependent protein from *Acipenser nacarii* (an Adriatic sturgeon), where highest expression was found in cartilage tissue [40]. The high number of Gla residues implements enhanced potential to bind to calcium or other mineral ions. In parallel, Surmann-Schmitt et al. identified the same protein named unique cartilage matrix-associated (UCMA) protein, which was also shown to be cartilage specific. Surmann-Schmitt et al. proposed that UCMA/GRP is a negative regulator of osteogenic differentiation as UCMA/GRP expression in chondrocytes which coincides with collagen and decreases when it reaches hypertrophy [41]. To further elucidate its role in skeletal tissue, a *Grp*^−/−^ mouse was generated. *Grp*^−/−^ mice showed normal cartilage development contrary to its expected role in osteogenic differentiation [42]. However, it was proposed by Eitzinger et al. that GRP has function in age associated maintenance of skeletal homeostasis based on its detection in adult cartilage. In the past decade, studies have reported the age associated effect of GRP in chronic diseases such as osteoarthritis (OA), another manifestation of ectopic calcification. Cavaco et al. showed that GRP expression was up-regulated throughout extracellular matrix (ECM) calcification, and treatment of chondrocytes with γ-carboxylated GRP resulted in the downregulation of inflammatory signals [43]. Further, Rafael et al. reported an association between uncarboxylated GRP (ucGRP) with OA cartilage. They used conformation specific antibodies against GRP and detected that γ-carboxylated GRP accumulates in control tissue, whereas ucGRP was mainly found at sites of ectopic calcification in OA-affected tissues [44]. Thereby indicating that γ-carboxylated GRP prevents ectopic calcification.

Furthermore, GRP has a role in vascular calcification process. Viegas et al. showed that both γ-carboxylated GRP and ucGRP accumulate at sites of mineral deposits in aorta and aortic valves [45]. They also showed that GRP is part of MGP–fetuin-A complex at the sites of valvular calcification. Under non-calcifying conditions, VSMC releases extracellular vesicles (EV) containing GRP, MGP, and fetuin-A, which depletes under calcifying conditions. This indicates that there is a tight crosstalk between these proteins in response to calcium. In concordance with these findings, recently Viegas et al. reported that calciprotein particles (CPP) and extracellular vesicles isolated from patients harboring chronic kidney disease (CKD) have lower levels of GRP and fetuin-A [46]. A major problem is that CKD patients have increased mineral maturation. In this study, CPPs and EVs from late-stage CKD patients (stage 5) were taken up by VSMCs in vitro, which promotes osteo-chondrogenic differentiation leading to vascular calcification. This phenotype was rescued in vitro when VSMCs were incubated with γ-carboxylated GRP. In addition, Willems et al. reported that GRP plays role in phosphate induced VSMC calcification, which is highly prevalent in CKD [47]. They showed higher expression of osteo/chondrogenic markers (BMP-2, Runx2, β-catenin, p-SMAD1/5/8, ALP, OCN) in VSMCs from *Grp*^−/−^ mice accompanied by enhanced mineralization and decreased expression of MGP. They also showed a direct interaction between GRP and BMP-2 signaling indicating role of GRP in regulation of osteo-chondrogenic differentiation. Until now, no human mutations in GRP have been described to cause a monogenic disorder. This plethora of new GRP related findings makes its mechanism of action an attractive topic to further understand its role in diseased condition.

### 2.3. Osteocalcin

Osteocalcin also known as bone gamma-carboxyglutamate protein (BGLAP) is another VKD proteins involved in the regulation of calcification [48]. It has the lowest number of only three Gla residues (17, 21, and 24) that undergoes γ-carboxylation. It is found in the extracellular matrix of bone and has function in bone mineralization. BGLAP further regulates osteoclast and osteoblast activity [28,49,50]. BGLAP also acts as a hormone in glucose homeostasis, cognition, and male fertility [51,52]. It was shown that γ-carboxylation of first Glu (Glu17) residue correlates with insulin resistance [53].

*Bglap*^−/−^ mice were born with normal mendelian frequency and were phenotypically normal at birth [48]. After six months they showed increased bone mass. This suggests that BGLAP deficiency might enhance osteoblast activity.

### 2.4. Non-Hemorrhagic Phenotypes of VKCFD1

Over the years, individual VKCFD1 cases were reported, who were diagnosed with non-hemorrhagic phenotypes as skin hyper-laxity, skeletal defects, and/or cardiac defects.

#### 2.4.1. Skeletal Defects

Congenital skeletal defects in VKCFD1 patients are diverse and affect different skeletal regions. Rost et al. were the first who reported a VKCFD1 patient with a slightly dysmorphic face along with reduced VKD clotting factor activities (GGCX:p.(G72_L124del);p.(R485P)) [9]. Dargouth et al. also reported a VKCFD1 patient with facial dysmorphia accompanied with stunted growth and developmental retardation (GGCX:p.(W157R);p.(D31N+T591K), Table 1) [14]. Furthermore, Watzka et al. reported six more patients showing skeletal anomalies and reduced bone mass density at early age [7]. Three patients were described to harbor midfacial hypoplasia or facial dysmorphism (GGCX:p.(p.G125R);p.(D534V), GGCX:p.(S284P);p.(W315X), and GGCX:p.(R83P);p.(R83P)). Another patient developed Chondrodysplasia punctata, a phenotype, which is characterized by calcified spots near the ends of bones and in cartilage (GGCX:p.(W157R);c.2085-5T>C). In addition, this patient was diagnosed with midfacial hypoplasia, a short neck, and with underdeveloped ear helix. Another patient with the genotype GGCX:p.(R485P);p.(W315X) was diagnosed with Chondrodysplasia punctata. Two VKCFD1 patients, who are siblings, were reported with the genotype GGCX:p.(R204C);p.(R204C). One of the sisters was diagnosed with midfacial hypoplasia, whereas the other sister has normal facial morphology in spite of harboring the same GGCX genotype.

Lastly, Tie et al. described one patient, who developed short distal phalanges of the fingers that was described as a known characteristic of Keutel syndrome (GGCX:p.(D153G);(R325Q+M174R)) [17]. Characteristics of Keutel syndrome further include cartilage calcification in the ears, nose, larynx, trachea, and ribs, and pulmonary artery stenoses.

#### 2.4.2. Skin Hyper-Laxity

Another observed mineralization defect in VKCFD1 patients are extreme skin hyperlaxity. Vanakker et al. described these skin defects as Pseudoxanthoma Elasticum (PXE)-like phenotype due to the similarities with PXE patients [8]. The classical PXE (OMIM no. #264800) is an autosomal recessive disorder characterized by yellowish papules or excessive skin folding on the neck and flexural areas accompanied by loss of vision and cardiovascular defects. PXE is caused by mutations in *ATP binding cassette subfamily C member 6* (*ABCC6*) gene that is an ATP dependent transmembrane transporter expressed mainly in the liver and kidney [54]. The skin defect in VKCFD1 patients also overlaps with Cutis laxa (OMIM no. #123700; #219100), another orphan disorder characterized by loose and sagging skin, which can be acquired or inherited. Mutations in *Elastin* and *Fibulin-5* gene have been reported to associate with patients developing cutis laxa [8]. Elastin and fibulin-5 proteins are present in the extracellular matrix and functions in maintaining the homeostasis of elastic fibers [55]. However, a detailed mechanism of how Elastin and fibulin-5 are involved in the development of cutis laxa is unknown.

Vanakker et al. were the first who described four VKCFD1 patients harboring a skin phenotype with yellowish papules or dot like depressions due to calcification of elastic fibers (GGCX:p.(R83W);p.(Q374X), GGCX:p.(G537A);p.(Q374X), GGCX:p.(V255M);p.(S300F), GGCX:p.(G558R);p.(F299S)) (Table 2). No mutation was found in the *ABCC6* gene. Ultrastructural findings of these patients showed fragmented and mineralized elastic fibers characterized by fine granular and bulky calcification in reticular dermis [8].

Since that time, more VKCFD1 patients have been described to have skin laxity. Li et al. reported two siblings with small yellowish papules forming large plaques at the age of 20 (GGCX:p.(R83W);p.(Q374X)) [56]. These patients were reported to have loose sagging skin near the axillary areas confirming features of PXE-like phenotype. Further analysis of skin biopsies demonstrated that ucMGP was the predominant form found in the mineralized areas in the patients’ skin.

Another family was reported, where two siblings developed loose and sagging skin, which primarily affect the neck and trunk region (GGCX:p.(V255M);p.(S300F)) [12].

Watzka et al. reported a patient to have mild skin laxity in the neck (GGCX:p.(H404P);p.(R485P)). Histological examination of a skin biopsy revealed fragmented elastic fibers with focal elastophagocytosis [7]. Okubo et al. reported a patient with a homozygous deletion mutation, GGCX:p.(S741LfsX100), who was diagnosed with a PXE-like phenotype. Histopathological examination revealed calcium deposits in elastic fibers and accumulation of collagen in the mid-dermis. Dermal fibroblasts from the patient showed higher expression of osteogenic markers like bone morphogenetic protein 6 and runt-related transcription factor 2. They also showed an increase in presence of ucMGP in patient dermal fibroblasts compared to controls [57]. Kariminejad et al. described 13 affected members from two families who developed PXE-like skin manifestations, loose sagging skin of the trunk and upper limbs, and retinitis pigmentosa in 10 members [58]. All affected family members were homozygous for the splice-site mutation GGCX:c.373+3G>T in the GGCX gene, which results in deletion of 53 aa (GGCX:p.(F73_G125del)). Heterozygous family members were unaffected.

Recently, another patient was reported with loose redundant skin, angioid streaks, and characteristic calcification of elastic structures in the mid dermis, who is compound heterozygous for: GGCX:c.200_201delTT and GGCX:p.(V255M) [11].

#### 2.4.3. Cardiac Defects

Broadly, cardiac abnormalities in VKCFD1 patients can be divided into congenital abnormalities and atherosclerosis. Three patients were reported with septal closure defects (GGCX:p.(W157R);p.(D31N+T591K), GGCX:p.(R83P);p.(R83P), and GGCX:p.(S284P);p.(W315X)). Two more patients with the genotype GGCX:p.(G72_L124del);p.(R485P) and GGCX:p.(W157R);c.2085-5T>C were reported with persistent ductus artheriosus Botalli that is characterized by defects in blood vessel closure [7,9,14]. Additionally, two patients were reported with pulmonary artery stenosis (GGCX:p.(V255M);p.(S300F) and GGCX:p.(R485P);p.(W315X)). Further two patients were reported with atherosclerosis at the ages of 47 and 46 (GGCX:p.(H404P);p.(R485P) and GGCX:p.(G537A);p.(Q374X)) [7,8].

### 2.5. Assays Determining γ-Carboxylation

All VKCFD1 patients have a clotting factor deficiency, which are routinely examined by measuring individual clotting factor activities from their blood samples once they display abnormal bleedings. Unfortunately, for most VKCFD1 patients’ examination stops with the identification of the reduced activities of the VKD clotting factors and the genetic confirmation of mutations in GGCX. Only few studies have measured the γ-carboxylation status of VKD non-hemostatic proteins such as BGLAP and MGP from the serum of VKCFD1 patients for research purpose [3,7,17]. Watzka et al. reported that vitamin K supplementation improved levels of γ-carboxylated BGLAP in the serum for three out of four VKCFD1 patients [7]. In addition, Tie et al. reported one patient, where vitamin K supplementation did not improve γ-carboxylation of MGP in plasma (GGCX:p.(D153G);(R325Q+M174R)) [17]. Some studies have also measured the γ-carboxylation status of MGP by immunohistochemical staining, which showed the presence of ucMGP in skin biopsies of VKCFD1 patients, who were reported with skin hyper-laxity ((GGCX:p.(V255M);p.(S300F), GGCX:p.(R83W);p.(Q374X), and GGCX:p.(S741LfsX100), GGCX:c.373+3G>T) [12,56,57]. Since not all the cases of VKCFD1 patients’ reports the γ-carboxylation status of non-hemorrhagic VKD proteins, the identification of causative proteins from the patient data is difficult. Therefore, in vitro assays, are useful to evaluate the effect of GGCX mutations on γ-carboxylation of VKD proteins.

The first in vitro studies of GGCX were based on either the incorporation of radioactive CO_2_ or by analyzing pentapeptide substrates, i.e., FLEEL or FLEEV. These assays helped in characterizing some mutations for example GGCX:p.(L394R) [59], which led to the identification of the Gla binding site. However, this short universal peptide assay has the limitation that whole VKD proteins cannot be analyzed. Recently, cellular systems have been established for in vitro analysis of GGCX mutations with respect to their ability to γ-carboxylate VKD proteins. Tie et al. have described a cell-based assay, where endogenous *GGCX* was knocked out by CRISPR-Cas9 genome editing in HEK293 cells [17]. In this study, γ-carboxylation activity of transfected *GGCX* mutations was measured for two stably co-expressed reporter proteins i.e., FIXgla-ProT and MGP-HPC_4_. The effect of three GGCX mutations on γ-carboxylation of FIXgla-ProT and MGP-HPC_4_ in dependence of different concentrations of Vitamin K_1_ were measured by sandwich ELISA. In this assay, FIXgla-ProT and MGP-HPC_4_ were captured by conformation-specific antibodies directed against either the FIX or MGP Gla domain and were detected by using antibodies against either prothrombin or HPC_4_ respectively. The same group has extended the number of analyzed GGCX mutations, where they have evaluated the effect of 45 reported GGCX mutations on γ-carboxylation of three reporter proteins (FIXgla-ProT, MGP-MBP, and BGLAP-His) [60]. Here, the tagged reporter proteins were captured by either anti-MBP or anti-His and were detected by protein specific anti-carboxylation antibody. Hao et al. showed that some GGCX mutations show differential effects on γ-carboxylation of VKD proteins. They concluded that for some mutations higher concentrations of vitamin K is required for improving the γ-carboxylation of extra-hepatic proteins MGP and BGLAP. Recently, our group has reported the effect of 20 GGCX missense mutations on γ-carboxylation of six different VKD proteins (GRP, MGP, BGLAP, GAS6, PRGP1, and TMG4) using a similar cell-based assay in HEK293 cells [61]. In this assay, *GGCX*^−/−^ HEK293T were transfected with bicistronic vectors containing cDNAs of GGCX wild-type or the mutant along with one VKD protein. Cell supernatants and pellets were collected to measure γ-carboxylation of secretory VKD proteins (GRP, MGP, BGLAP, GAS6) and transmembrane VKD proteins (PRGP1 and TMG4) respectively. γ-Carboxylation of VKD proteins were measured by sandwich ELISA, where the VKD protein was captured by an anti-myc antibody followed by γ-carboxylation detection by a monoclonal antibody directed against human Gla residues for all analyzed VKD proteins. We correlated the γ-carboxylation values obtained from this in vitro assay with the genotype of VKCFD1 patients harboring a non-hemorrhagic phenotype. By this we mainly identified that GGCX mutations with markedly reduced ability to γ-carboxylate GRP in our in vitro assay are reported in VKCFD1 patients that developed skin hyperlaxity.

Overall, the data of Hao et al. and Ghosh et al. show similar results for most mutations, although there are discrepancies in γ-carboxylation values for some mutations. This could be because different tags were used in these studies to capture the VKD proteins in the ELISA and/or due to the different expression systems that were used. Simultaneous transient expression of GGCX and the VKD reporter in one cell line was used in the study of Ghosh el al., whereas Hao et al. used transient expression of GGCX in different cell lines that stably over-express the reporter VKD protein. However, altogether, these cellular in vitro studies have helped to understand the association of non-hemostatic VKD proteins to non-hemorrhagic phenotypes in VKCFD1 patients that is discussed in the next section.

### 2.6. Potential Roles of ucMGP and ucGRP in the Development of Mineralization Defects in VKCFD1 Patients

#### 2.6.1. Skeletal Defects

Due to the emerging knowledge on MGP and the association of loss-of-function mutations with Keutel syndrome, it was anticipated that undercarboxylated MGP is the major determinant in the development of skeletal defects in VKCFD1 patients. In line with this assumption, Tie et al. reported a VKCFD1 patient, who developed a Keutel syndrome-like phenotype, where the pathogenic mutations GGCX:p.(M174R+R325Q) and p.(D153G) showed extremely reduced ability to γ-carboxylate MGP in their in vitro study [17]. However, not all GGCX mutations, which were reported in the other VKCFD1 patients harboring skeletal defects show markedly, reduced ability to γ-carboxylate MGP in the study of Hao et al. and Ghosh et al. (Table 1, for example GGCX:p.(S284P);p.(W315X)). On the other hand, there are VKCFD1 patients, who did not develop skeletal defects in spite of harboring GGCX mutations that resulted in extremely diminished ability to γ-carboxylate MGP in vitro. One example is the patient with the genotype GGCX:p.(G558R);p.(F299S), who does not display skeletal defects, although in vitro γ-carboxylation measurements of MGP were nearly zero for both GGCX mutations by Hao et al. and Ghosh et al. [60,61].

Therefore, results from both studies eliminate the hypothesis that under-carboxylated MGP is the sole causative determinant for developing skeletal defects in patients. We assume that the development of skeletal defects is attributed to various factors as for example nutritional uptake of the mother during pregnancy, polymorphisms in VKORC1 gene, or other epigenetic factors. Maternal nutrition during pregnancy is crucial for the normal development of the offspring. Since vitamin K is supplied by food, insufficient intrauterine transfer of vitamin K or insufficient transfer of γ-carboxylated VKD proteins might lead to skeletal defects in the offspring. This hypothesis is supported by one case report harboring the genotype GGCX:p.(S284P);p.(W315X) [7], where the mother of the patient developed hyperemesis gravidarum with a weight loss of 7 kgs in the first trimester. GGCX:p.(S284P) showed 134% γ-carboxylation for MGP in vitro by Hao et al. and 102% by our group. However, the offspring developed skeletal defects, despite having theoretically sufficient amounts of γ-carboxylated MGP. This case report shows the importance of nutritional uptake during pregnancy.

Apart from nutritional uptake, the availability of substrate (reduced vitamin K) is of importance and might be also affected by the promoter single nucleotide polymorphism (SNP) in VKORC1 gene NG_011564.1:g.3588G>A (VKORC1:c.-1639 G>A). It was shown by Rieder et al. that the VKORC1:c.-1639 AA polymorphism in the promoter of VKORC1 is associated with 66% reduced mRNA expression compared to VKORC1:c.-1639 GG [62]. Thus, availability of KH_2_ could be largely influenced, which has no effect on healthy individuals but in VKCFD1 patients. Interestingly, two patients with midfacial hypoplasia with the genotype GGCX:p.(S284P);p.(W315X) and GGCX:p.(R83P);p.(R83P) are homozygous for the AA polymorphism in the VKORC1 promoter. Therefore, the reduced VKORC1 expression might correlate with reduced availability of KH_2_, which might affect the skeletal development. This is also evident by the siblings that harbor the same GGCX genotype (GGCX:p.(R204C);p.(R204C)), where only one of them developed midfacial hypoplasia. The sister who developed midfacial hypoplasia is heterozygous for the VKORC1:c.-1639 polymorphism, which is associated with 33% reduced VKORC1 mRNA compared to the other sister, who is homozygous for VKORC1:c.-1639 GG, which is associated with 100 % VKORC1 mRNA. Thus, the cumulative effect of GGCX mutations in the offspring along with either reduced VKORC1 expression in the offspring; or nutritional uptake during pregnancy cause congenital skeletal defects in rare cases of VKCFD1 patients (Figure 1).

#### 2.6.2. Skin Hyper-Laxity

It has been a decade since the PXE-like phenotype has been described in VKCFD1 patients but the pathomechanism has not been yet identified. We have recently reported that under-carboxylation of GRP by specific GGCX mutations are associated with skin hyper-laxity in VKCFD1 patients [61]. Our in vitro assay revealed γ-carboxylation efficiency of 18–32% for GRP by GGCX mutations that were reported in patients harboring skin hyperlaxity (Table 2, Figure 1). Viegas et al. have showed that GRP is expressed in the skin in humans and rats, which is in concordance with our findings that under-carboxylation of GRP might cause a skin phenotype [63]. They also showed that GRP is highly expressed and accumulates in small blood vessels and capillaries that irrigate the skin and in fibroblasts of both papillary and reticular dermis in hair follicles and sebaceous glands. Therefore, levels of GRP (under-carboxylated and γ-carboxylated) should be analyzed in skin biopsies ofVKCFD1 patients with skin hyper-laxity.

With respect to MGP the analysis of Hao et al. showed 0–200% of γ-carboxylation for MGP for GGCX mutations that were reported in patients with skin hyper-laxity. The γ-carboxylation values for MGP of our study were in the range of 3–102% (Table 2). Therefore, MGP can be excluded as a sole causative VKD protein for developing skin hyper-laxity. However, ucGRP might negatively regulate MGP levels, since some studies have shown the presence of ucMGP in skin biopsies of VKCFD1 patients.

#### 2.6.3. Cardiac Abnormalities

The GGCX mutations identified in patients with congenital cardiac abnormalities showed 0–134% of γ-carboxylated MGP levels by the in vitro analysis of Hao et al. In concordance with their findings, our study showed for the same mutations 3–100% γ-carboxylation for MGP and 1–80% for GRP (Table 1 and Table 2). Interestingly, all patients with congenital cardiac abnormalities were additionally reported with skeletal dysmorphologies (Figure 1) except for the patient harboring the mutations GGCX:p.(V255M);p.(S300F).

Other VKCFD1 patients, who developed atherosclerosis in adulthood were also reported with mild to severe skin hyper-laxity. The γ-carboxylated MGP levels by these GGCX mutations were 37–94% by the analysis of Hao et al. and 9–102% by our study (Table 1 and Table 2). So, no trend of under-carboxylation of MGP was observed from both studies. However, the γ-carboxylated GRP levels were reduced to 18–58% by GGCX mutations reported in patients with atherosclerosis. Imbalance of γ-carboxylated MGP and GRP in cardiac diseases as well as their interaction with each other are well-investigated [33,45]. A recent study by Willems et al. demonstrated that VSMCs isolated from *Grp*^−/−^ mice showed decreased expression of MGP. Therefore, there is a possibility that reduced levels of γ-carboxylated GRP might negatively regulate MGP levels. However, more studies are needed that investigate the role of under-carboxylated MGP and GRP in developing cardiac abnormalities in VKCFD1 patients but also in healthy individuals especially under challenging/calcifying conditions.

With respect to BGLAP no clear trend was observed in the in vitro data when correlated with the genotypes of VKCFD1 patients with non-hemorrhagic phenotypes.

### 2.7. Treatment of Non-Hemorrhagic Phenotypes in VKCFD1

VKCFD1 patients are treated with high doses of phylloquinone/vitamin K_1_ (10 mg/day) for increasing the clotting factor activities, but it is unclear whether life-long treatment with vitamin K_1_ is sufficient to rescue age associated non-hemorrhagic phenotypes. To prevent congenital skeletal defects in the offspring, treatment of vitamin K during pregnancy might be useful. However, reports for those cases are missing.

The age of onset for skin hyper-laxity was reported to be around 18 years and at later age [19]. Therefore, early-stage monitoring and vitamin K enriched diet might prevent or postpone the onset of skin laxity for the patients harboring *GGCX* mutations that effect γ-carboxylation of GRP mildly. However, our recent in vitro study showed that *GGCX* mutations, reported in VKCFD1 patients diagnosed with a PXE-like phenotype, do not respond with increasing levels of γ-carboxylated GRP after treatment with increasing vitamin K concentrations. Therefore, theoretically VKCFD1 patients might also not show greatly enhanced γ-carboxylated GRP values under vitamin K enriched diet. In this case, other treatment options than only vitamin K administration are needed to rescue the development of skin folding and hyper-laxity. Viegas et al. have proposed a treatment strategy to inhibit vascular calcification by a nanotechnology-based method, where human γ-carboxylated GRP is loaded into extracellular vesicles [64]. This system was capable of reducing extracellular matrix calcium deposits in calcifying VSMCs in vitro. However, it remains to be elucidated whether this innovative system will prevent skin hyperlaxity.

For age associated cardiac defects or reduced bone mass density, the genotype might determine the outcome of vitamin K treatment since studies of Watzka et al. and Tie el al. showed that for certain genotypes MGP or BGLAP levels are not increasing in patients by vitamin K supplementation [3,7,17]. Therefore, for certain genotypes additional application of γ-carboxylated GRP might be useful.

Another issue is that VKCFD1 patients are treated with vitamin K_1_ in order to correct the bleeding phenotype. However, studies have shown that vitamin K_2_ has a longer half-life in the circulation and thus is more available for the extra-hepatic tissues [65,66]. Therefore, a combination treatment with K_1_ and K_2_ might be a good strategy for treating the hepatic coagulation deficiency and extra-hepatic abnormalities, respectively.

## 3. Conclusions

With the advancement of genotyping techniques and increasing knowledge, more VKCFD1 patients with non-hemorrhagic phenotypes have been reported. To further understand the diversity of these phenotypes, follow up reports of these patients are crucial. Since recent studies indicate the involvement of GRP and MGP in calcification defects, where for example GRP serves as marker for kidney dysfunction in CKD patients [PMID: 32120910], levels of under-carboxylated and γ-carboxylated GRP along with MGP and BGLAP should be used as a biomarker in VKCFD1 patients as well. It would be interesting to measure the values before and after vitamin K treatment and to correlate the values with the severity of non-hemorrhagic phenotypes in VKCFD1 patients. We also recommend to genotype VKCFD1 patients additionally for the VKORC1 promotor polymorphism since there is evidence that reduced VKORC1 expression could affect the severity of VKCFD1 phenotypes. Functional studies by new cell-based systems have helped in understanding the effect of GGCX mutations on γ-carboxylation of non-hemorrhagic VKD proteins. However, as these VKD proteins show variable expression in different tissue, where the absorption or type of vitamin K is different, tissue and patient specific studies are essential to further understand the mechanism of action.

Overall, further research is needed to understand the mechanism of inhibition of calcification in extra hepatic tissue by GRP and MGP, which will pave the way for personalized therapy.

## Figures and Tables

**Figure 1 ijms-23-00798-f001:**
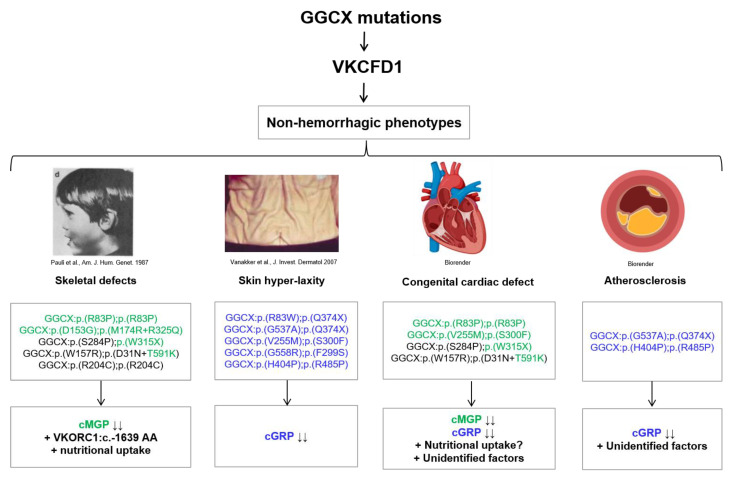
Evaluation of under-carboxylated MGP and GRP in the development of non-hemostatic VKCFD1 phenotypes. The schematic shows the genotype of VKCFD1 patients that were reported with skin hyper-laxity, skeletal dysmorphologies, congenital cardiac abnormalities and/or atherosclerosis. GGCX mutations that showed reduced ability to carboxylate MGP or GRP in vitro are highlighted in green (for MGP) or blue color (GRP). GGCX mutations in black color showed γ-carboxylation levels above 50% in vitro. Multiple factors could be associated with skeletal dysmorphologies such as under-carboxylated MGP, VKORC1:c.-1639 AA polymorphism and nutritional uptake by the mother during pregnancy. The biallelic reduction to γ-carboxylate GRP is associated with the PXE-like phenotype in VKCFD1 patients. Congenital cardiac abnormalities are associated with multiple factors such as ucMGP, nutritional uptake during pregnancy or other not yet identified factors. In the two patients with subclinical atherosclerosis low levels of γ-carboxylated GRP could be the reason for developing this phenotype.

**Table 1 ijms-23-00798-t001:** Patient and in vitro data with respect to GGCX mutations causing skeletal defects in VKCFD1 patients’ genes, age, and non-hemorrhagic phenotypes and in vitro γ-carboxylation levels of MGP and GRP measured by cell based assays reported by Hao et al. and Ghosh et al. NA, not available; ND, not determined.

Genotype (Allele 1 Allele 2)	VKORC1 c.-1639 (VKOR Activity)	Age	Non-Hemorrhagic Phenotypes	Refs.	In Vitro γ-Carboxylation from Cell Based Assay		
					γ-Carboxylated MGP	γ-Carboxylated MGP	γ-Carboxylated GRP
					Hao et al.	Ghosh et al.	Ghosh et al.
R83P R83P	AA (33%)	3 years	Facial Dysmorphism + Septal defect	Watzka et al., 2014	76% 76%	27% 27%	13% 13%
D153G M174R + (R325Q)	NA	4 months	Keutel syndrome like phenotype	Tie et al., 2016	39% 0%	38% 1%	93% 1%
W157R (D31N) + T591K	NA	11 years	Developmental delay and stunted growth + Septal Defect	Dargouth et al., 2006	34% (108%) + 9%	58% 24%	44% 32%
R204C R204C	GA (66%)	11 years	Midfacial hypoplasia	Watzka et al., 2014	20% 20%	55% 55%	67% 67%
R204C R204C	GG (100%)	14 years	Face morphology normal	Watzka et al., 2014	20% 20%	55% 55%	67% 67%
S284P W315X	AA	13 years	Midfacial hypoplasia + Septal defect	Watzka et al., 2014	134% 0%	100% ND	80% ND
R485P W315X	GA	14 years	Chondrodysplasia punctate and pulmonary arterial stenosis	Watzka et al., 2014	37% 0%	87% ND	58% ND
G125R D534V	AA	5 years	Mild midfacial hypoplasia	Watzka et al., 2014	0% 43%	ND ND	ND ND
G72_L124del R485P	NA	1 year	Midfacial hypoplasia and persistent ductus artheriosus Botalli	Rost et al., 2004	ND 37%	ND 87%	ND 58%

**Table 2 ijms-23-00798-t002:** Patient and in vitro data with respect to GGCX mutations causing skin hyper-laxity in VKCFD1 patients. Table with previously reported patients data (GGCX genotype, promotor polymorphism in VKORC1 gene, age and non-hemorrhagic phenotypes) and in vitro γ-carboxylation levels of MGP and GRP measured by cell based assays reported by Hao et al. and Ghosh et al.. NA, not available; ND, not determined.

Genotype (Allele 1 Allele 2)	VKORC1 c.-1639	Age	Non-Hemorrhagic Phenotypes	Refs.	In Vitro γ-Carboxylation from Cell Based Assay		
					γ-Carboxylated MGP	γ-Carboxylated MGP	γ-Carboxylated GRP
					Hao et al.	Ghosh et al.	Ghosh et al.
R83W Q374X	NA	46 years	Skin hyper-laxity	Li et al., 2009	81% 0%	32% ND	26% ND
V255M S300F	NA	16 years	Skin hyper-laxity + peripheral pulmonary artery stenosis	Li, Grange et al., 2009	163% 21%	10% 3%	31% 1%
G558R F299S	NA	40 years	Skin hyper-laxity	Vanakker et al., 2007	13% 0%	18% 1%	32% 0%
H404P R485P	GG (100%)	47 years	Mild skin symptom + Calcified peripheral arteries	Watzka et al., 2014	78% 37%	9% 87%	35% 58%
G537A Q374X	NA	46 years, 44 years	Skin hyper-laxity + Atherosclerosis	Vanakker et al., 2007	94% 0%	102% ND	18% ND

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
