# Peer review of "The Role of GRP and MGP in the Development of Non-Hemorrhagic VKCFD1 Phenotypes"

_ijms, 2022, doi:10.3390/ijms23020798_

Round 1
Reviewer 1 Report
The review is well written and comprehensive
I have no comments
Author Response
We would like to thank reviewer 1 for his time, for reviewing our article and his positive response.
Reviewer 2 Report
In this review of Ghosh et al. the authors describe the heterogeneous phenotype of VKCFD1 mutations focusing on the role of matrix Gla protein (MGP) and Gla-rich protein (GRP). They have summarized the recent findings on the function of GRP and MGP and how these proteins influence the development of non-hemorrhagic phenotypes in VKCFD1 patients. MGP inhibits calcification and the osteogenic trans-differentiation of vascular smooth muscle cells is regulated by MGP’s interaction with bone morphogenic protein-2. The rare Keutel syndrome is the only one monogenic disorder associated with mutations in non-haemostatic vitamin K dependent proteins. GRP is a negative regulator of osteogenic differentiation and it is suggested to have function in age associated maintenance of skeletal homeostasis. GRP also has a role in vascular calcification processes.
The review collects all the relevant pieces of information what are known today about these proteins and VKCFD1-related non-haemostatic diseases/symptoms and cites more than 60 papers. It is a useful reading for those dealing with rare diseases. I only have some minor issues.
- It would be interesting to add some additional data interesting from the point of view of laboratory medicine. Are there any recommendations about the usefulness of measurement of MGP and GRP levels for example in different conditions? The authors mention lower levels of GRP in chronic kidney disease. Could it be a prognostic marker of such diseases?
- I think the sentence “Altogether, these studies show that VKD non-hemostatic proteins- MGP and GRP plays a significant role in preventing pathological conditions.” is too indefinite. You may omit this or make it more specific.
- The method of the authors by which in vitro γ-carboxylation levels of MGP and GRP is measured in the cell based assays should be described here in more details as they provide important laboratory data. Could you please comment the discrepancy between Hao’s method and yours and give a recommendation how to measure γ-carboxylation in the research laboratory?
I agree with the conclusions drawn based on the available literature data and agree with the initiative for more functional studies to find the optimal treatment for patients.
Author Response
First of all we would like to thank reviewer 2 for reviewing our article and his suggestions, which improved the quality of the manuscript a lot.
Below there is a point-by-point response:
- It would be interesting to add some additional data interesting from the point of view of laboratory medicine. Are there any recommendations about the usefulness of measurement of MGP and GRP levels for example in different conditions? The authors mention lower levels of GRP in chronic kidney disease. Could it be a prognostic marker of such diseases?
Answer:
We agree with the reviewer. To our knowledge MGP and GRP are currently not routinely measured, just for research purpose only. However, we have included in our conclusion that measuring levels of GRP and MGP would be very useful as biomarkers since other studies have already shown that decreased levels of GRP correlate with kidney dysfunction in CKD patients. We have also included the study of Silva et al. now into our review (PMID: 32120910).
- I think the sentence “Altogether, these studies show that VKD non-hemostatic proteins- MGP and GRP plays a significant role in preventing pathological conditions.” is too indefinite. You may omit this or make it more specific.
Answer:
We agree with the reviewer and have omitted the above-mentioned sentence.
- The method of the authors by which in vitro γ-carboxylation levels of MGP and GRP is measured in the cell based assays should be described here in more details as they provide important laboratory data. Could you please comment the discrepancy between Hao’s method and yours and give a recommendation how to measure γ-carboxylation in the research laboratory?
Answer:
We have added more details on the different cell based assays in pages 7 and 8. We have discussed the discrepancies in the carboxylation values for specific GGCX mutation between Hao et al. and our group as well. Briefly, these differences could be due to the different antibodies and cell lines that were used.
Honestly, we cannot give a recommendation, which assay should be used f.e for newly identified mutations. We think that these assays are the best that are currently available to measure mutation specific carboxylation defects in the laboratory. However, we think that the next essential step are patient derived samples, where GGCX and carboxylation of VKD proteins can be measured on endogenous level. We have pointed out this also in the conclusions.